# Combination of Different Fecal Immunochemical Tests in Colorectal Cancer Screening: Any Gain in Diagnostic Performance?

**DOI:** 10.3390/cancers11010120

**Published:** 2019-01-20

**Authors:** Anton Gies, Katarina Cuk, Petra Schrotz-King, Hermann Brenner

**Affiliations:** 1Division of Preventive Oncology, German Cancer Research Center (DKFZ) and National Center for Tumor Diseases (NCT), 69120 Heidelberg, Germany; anton.gies@nct-heidelberg.de (A.G.); petra.schrotz-king@nct-heidelberg.de (P.S.-K.); 2Medical Faculty Heidelberg, University of Heidelberg, 69120 Heidelberg, Germany; 3Division of Clinical Epidemiology and Aging Research, German Cancer Research Center (DKFZ), 69120 Heidelberg, Germany; k.cuk@dkfz.de; 4German Cancer Consortium (DKTK), German Cancer Research Center (DKFZ), 69120 Heidelberg, Germany

**Keywords:** colon cancer, advanced neoplasia, fecal occult blood test, early detection, prevention

## Abstract

A variety of fecal immunochemical tests (FITs) are used for colorectal cancer screening. FIT performance could be improved further. It is unclear, whether the combination of different FITs with different analytical characteristics (such as, different antibodies for the detection of fecal hemoglobin) can yield a better diagnostic performance. Fecal samples were obtained from 2042 participants of screening colonoscopy. All participants with advanced neoplasm (AN, colorectal cancer (*n* = 16) or advanced adenoma (*n* = 200)) and 300 randomly selected participants without AN were included. Nine quantitative FITs were evaluated simultaneously. Sensitivity and specificity was calculated for single tests (*n* = 9) and for their pairwise test combinations (*n* = 36) (requiring either both FITs (P++) or at least one FIT (P+) to be positive for defining a positive test result). Mean age of the participants (*n* = 516) was 63 (range: 50–79) years and 56% were men. At cutoffs yielding a specificity of 96.7% for single FITs, the median gain in specificity by P++ combination was +1.0%, whereas the median loss in sensitivity for AN was −4.2%. For P+ combination the median gain in sensitivity for AN was +2.8%, at a prize of median loss of −1.0% of specificity. Combinations of different FITs do not yield any relevant gain in diagnostic performance.

## 1. Introduction

Colorectal cancer (CRC) accounts worldwide for approximately 1.8 million new cases and 0.9 million deaths in 2018 [1]. Randomized trials have demonstrated that screening with guaiac-based fecal occult blood tests (gFOBTs) can reduce CRC mortality by up to 30% [2,3,4]. Meanwhile, fecal immunochemical tests (FITs) for hemoglobin (Hb) are widely recommended and used for CRC screening in many countries [5,6,7,8,9], as they have been shown to yield substantially better diagnostic performance [10,11,12] and significantly higher participation rates [13,14] in direct comparisons with gFOBTs. However, the sensitivity of FITs for detection of advanced neoplasm (AN; i.e., CRC or advanced adenoma (AA)) is rather low (reported range: 9–60%) [15] and could be improved further.

Former studies [11,16,17,18] have assessed whether multiple FIT measurements and their combinations based on the collection of multiple stool samples from different bowel movements can improve diagnostic performance. Even though they found an increase in sensitivity, this went along with a reduced specificity, or vice versa; and the area under the curve for AN was not significantly different in comparison to a one FIT sample regime. However, all these former evaluations were based on combinations of the same FIT brand (OC Sensor, Eiken Chemical, Tokyo, Japan). Meanwhile, a large variety of different FITs from diverse manufactures with different analytical characteristics (such as different antibodies for hemoglobin detection, different analytical reading techniques) have entered the market [15,19,20,21,22]. Therefore, it is essentially unclear whether a combination of different FITs with different analytical characteristics could yield a relevant gain in diagnostic performance.

We aimed to explore the potential for enhancing diagnostic performance by pairwise combinations of nine different quantitative FITs based on stool samples obtained from a large cohort of participants of the German screening colonoscopy program.

## 2. Results

### 2.1. Study Population

A total of 2042 participants of screening colonoscopy (Begleitende Evaluierung innovativer Testverfahren zur Darmkrebsfrüherkennung (BLITZ) study, screening setting), who provided stool samples in 60 mL containers were eligible for this project. After excluding 375 participants due to exclusion criteria shown in Figure 1 (left side), 1667 participants were left to choose from. All eligible participants with CRC (*n* = 16) or AA (*n* = 200) and 300 randomly selected individuals without CRC and AA were included. A slight majority of the participants were men (56%) and mean age was 63.2 years.

Due to the low number of screening CRC cases (*n* = 16), an ancillary group of CRC cases (*n* = 50) from a clinical setting (Darmkrebs: Chancen der Verhütung durch Screening (DACHS) plus study) were included. From a total of 184 clinical CRC patients, 90 study participants were excluded, due to exclusion criteria shown in Figure 1 (right side). From the remaining 94 individuals, all eligible participants with screen-detected CRC were included (*n* = 27), and 23 randomly-selected participants, who were detected otherwise, were additionally added. Thirty individuals (60%) were male and mean age was 65.8 years.

### 2.2. Sensitivity and Specificity

Figure 2 shows sensitivities and specificities for detecting AN in the screening setting (BLITZ study) for the single tests and for their pairwise combinations (if both tests are positive (P++) or if at least one test is positive (P+)) Figure 2a at original cutoffs (range: 2–17 µg Hb/g feces), Figure 2b at a uniform cutoff (15 µg Hb/g feces) and at cutoffs adjusted to yield the same specificity (97% Figure 2c and 93% Figure 2d, respectively) for the single tests.

At original cutoffs, median sensitivity for AN was 34.7% (range: 21.8–46.3%) across all nine FITs and median specificity for participants without AN was 91.3% (range: 85.7–97.7%). Pairwise combinations with both tests to be positive (P++) led to a strong reduction of the sensitivity of the respective tests by a median of −13.7% units (range: −25.5 to −1.4% units), whereas the specificity increased just slightly by a median of 0.7% units (range: 0.0 to 4.7% units). Combinations of tests with at least one test to be positive (P+) resulted in a slight increase of the sensitivity by a median of 1.2% units (range: 0.0 to 5.1% units), at a prize of a stronger median loss of −6.7% units (range: −12.7 to −1.3% units) of the specificity.

At a uniform cutoff, the median sensitivity and specificity for AN was 21.8% (range: 16.2–34.3%) and 96.3% (range: 94.0–98.7%), respectively. For P++ combinations, the sensitivity decreased by a median of −7.0% units (range: −18.5 to −1.9% units) and the specificity increased only marginally by a median of 0.7% units (range: 0.0 to 1.7% units). For P+ combinations, the sensitivity increased only slightly by a median of 1.4% units (range: 0.0 to 4.2% units) and this went along with a similarly strong reduction of the specificity by a median of –2.3% units (range: −4.7 to −0.7% units).

In order to enhance the comparability of diagnostic performance of the single tests with the test combinations, sensitivities were calculated at cutoffs adjusted to yield the same specificity of 96.7% (range of applied cutoffs: 6–30 µg/g) and 93.0% (range of applied cutoffs: 2–13 µg/g), respectively. At 96.7% specificity, the sensitivities were very similar across all nine FITs, ranging from 21.3% to 23.6% (median: 22.7%). For P++ combinations, the specificity increased slightly by a median of 1.0% unit (range: 0.3 to 2.3% units), however, the sensitivity decreased much stronger, with a median reduction of −4.2% units (range: −5.6 to −1.4% units). Whereas for P+ combinations, the specificity declined slightly by a median of −1.0% unit (range: −2.3 to −0.3% units) and the sensitivity increased only marginally by a median of 2.8% units (range: 0.9 to 4.7% units). Similar observations were made a cutoffs adjusted to yield 93% specificity.

In addition, Table 1 and Table 2 presents summary results on the sensitivities for CRC (separately for screening (*n* = 16) and clinical (*n* = 50) CRC cases) as well as for AA cases (*n* = 200) and the absolute differences in percent units between the respective test combinations and the single tests with the higher sensitivity and specificity, respectively.

### 2.3. Area Under the Curve (AUC)

The single AUCs of the FITs for detection of AN in the screening setting ranged from 59.5% (QuikRead go iFOBT, Orion Diagnostica, Espoo, Finland) to 72.1% (IDK Hb ELISA, Immundiagnostik, Bensheim, Germany), with a median AUC of 68.9% (QuantOn Hem, Immundiagnostik, Bensheim, Germany). Pairwise combinations yielded combined AUCs for AN between 63.0% (QuikRead go iFOBT + SENTiFIT-FOB Gold) and 72.8% (IDK Hb ELISA + RIDASCREEN Hb). The absolute gain in AUC for AN between the best single test (IDK Hb ELISA) and the best pairwise test combination (IDK Hb ELISA + RIDASCREEN Hb) was only +0.7% units, revealing no meaningful gain in diagnostic performance by pairwise test combinations.

Even when all nine FITs were combined together, no gain in diagnostic performance was observed. The AUC for AN by combining all nine FITs was 71.6%, which was even below the single AUC value of IDK Hb ELISA.

### 2.4. Correlation Analyses

#### 2.4.1. Spearman Correlation between Test Measurements

Spearman rank coefficients (r_s_) between the quantitative fecal Hb measurements across all 36 two-test comparisons ranged from 0.81 to 0.98 (median: 0.89) for CRC (*n* = 16), from 0.60 to 0.95 (median: 0.78) for AA (*n* = 200), from 0.22 to 0.81 (median: 0.44) for participants without CRC and AA (*n* = 300), and from 0.51 to 0.90 (median: 0.66) for all study participants of screening colonoscopy (*n* = 516) (Table 3).

#### 2.4.2. Cohen’s Kappa for Agreement of Test Classification

After adjusting the original cutoffs to yield an equal specificity of 96.7%, the median kappa coefficients across all 36 two-test comparisons increased from the original cutoffs to the adjusted cutoffs from 0.67 (range: 0.56–1.00) to 0.86 (range: 0.71–1.00) for CRC (*n* = 16), from 0.64 (range: 0.40–0.92) to 0.78 (range: 0.66–0.92) for AA (*n* = 200), from 0.45 (range: 0.15–0.85) to 0.69 (range: 0.28–0.90) for participants without CRC and AA (*n* = 300), and from 0.64 (range: 0.41–0.87) to 0.81 (range: 0.66–0.91) for all study participants of screening colonoscopy (*n* = 516) (Table 4).

## 3. Discussion

This is the first study assessing the potential of improving the diagnostic performance of FITs by combining nine different FITs with different analytical characteristics, based on stool samples obtained from a large cohort of average-risk participants of screening colonoscopy.

We found that the nine different FITs correlated strongly across each other, although the tests were different in their analytical characteristics (e.g., different antibodies for detection of human Hb, different analytical reading techniques). The different FITs obtained very similar sensitivities and specificities, and detected in majority the same CRC and AA cases, without any relevant gain in test performance by combining two different FITs. Even the combination of all nine FITs together yielded no gain in diagnostic performance.

Previous colonoscopy-controlled studies [11,16,17,18] have investigated whether the diagnostic performance could be improved by combining multiple FIT samples, collected from different bowel movements, but these studies were based on using the same FIT (OC Sensor, Eiken Chemical, Tokyo, Japan): Oort et al. [16] evaluated the diagnostic performance of 2 FIT samples and found a slight increase in sensitivity, but this went along with a reduced specificity, or vice versa, when either at least one test or both tests were required to be positive for defining a positive test result. Similarly, Hernandez et al. [17] and Liles et al. [18] combined two FIT samples by using the higher fecal Hb measurement of both FIT samples for test interpretation and observed no relevant gain in test performance in comparison to one FIT sample only. In another study from South Korea, Park et al. [11] combined up to three FIT samples by taking the highest test result into test interpretation, but the improvement in sensitivity went along again with a reduction in specificity, and AUCs for AN were not significantly different. However, because in all these studies the FIT combinations were based on the same FIT (OC Sensor, Eiken Chemical, Tokyo, Japan) with the same analytical characteristics, it was essentially unclear whether a combination of FITs with different analytical characteristics could yield a relevant improvement in diagnostic performance.

In a previous analysis we evaluated and directly compared for the first time, the individual diagnostic performance of these nine quantitative FITs, and found apparent large differences regarding the diagnostic performance at original cutoff values recommended by the manufacturers (range: 2–17 µg Hb/g feces) [22]. However, after adjusting the cutoffs to yield equal specificities (here: 99%, 97% and 93%, respectively), very similar sensitivities and almost identical positivity rates (here: ≈3%, ≈6% and ≈11%, respectively) were observed. However, the single FIT measurements varied widely across the tests and it was necessary to set partly very different cutoff values (range: 18–53 µg Hb/g feces, 6–30 µg Hb/g feces, and 2–12 µg Hb/g feces, respectively) to yield the same specificity at almost identical positivity rates among all nine tests. These variations between the tests reflect the different analytical characteristics across the tests, for example, the different analytical reading techniques, the different antibodies for Hb detection or their different affinities to bind (partly) degraded Hb variants. In this study, we therefore aimed to correlate and combine these nine previously evaluated FITs in order to investigate whether the diagnostic performance of single FITs could be improved by combining different FITs with different analytical characteristics. Interestingly, we found that the different FITs correlate strongly across each other, although they have different analytical characteristics. They detected in majority the same CRC and AA cases, and no relevant gain in diagnostic performance was observed across all test combinations. Furthermore, with cutoff values adjusted to yield the same specificity or overall positivity rate [15,22], very similar sensitivities can be achieved with a variety of different FITs.

Specific strengths of our study are the first time combination of a large number of different quantitative FITs with different analytical characteristics, based on exactly the same stool samples obtained from average-risk participants of screening colonoscopy. The stool samples were collected before starting bowel preparation for colonoscopy. Screening colonoscopy, which is the current diagnostic gold-standard was performed independently of the FIT and was used as the reference standard to evaluate the diagnostic performance. The study design essentially precluded variations, which could have occurred from different study populations (e.g., case numbers; age [23,24,25], sex [23,24,25,26,27], and stage [11,22,28,29,30] distribution) or from different pre-analytical sample handlings [31] between the FITs: Stool samples were collected in exactly the same manner and additional mixing of the stool samples, before filling the fecal sampling tubes, was performed to account for potential heterogeneity in Hb concentration within the same bowel movement [32] as an additional source of variation of test results between the different FITs. Furthermore, the stool-filled FIT tubes were stored under very similar ambient temperatures (range: 20.0 °C to 24.0 °C) and were evaluated in parallel on the same day to rule out differences in pre-analytical sample handling [31] as another source of test variation.

However, our study has also some limitations. The stool samples were collected by the study participants in stool containers (60 mL) and stored frozen at −80 °C over several years prior to filling the special fecal sampling tubes for test analysis, instead of directly collecting the stool samples with the recommended fecal sampling tubes provided by the manufacturers. However, as it is difficult to imagine that study participants would be willing and able to collect nine FIT samples in parallel using nine different fecal sampling tubes (each with different fecal sampling instructions and FIT-specific peculiarities), this was probably the only way to realize a study with so many simultaneously evaluated FITs. Nevertheless, the original FIT tubes provided by the manufacturers were used to collect a defined amount of stool after thawing the stool containers, and the stool specimens were stored in their respective preservative buffer from sampling until test evaluation. Furthermore, the stool-filled FIT tubes were vortexed so that the stool could move out of the notches of the stick and disperse completely into the buffer of the tube to ensure optimal Hb stabilization during the study [31]. In addition, we found in a previous examination based on one of the nine tests (SENTiFIT-FOB Gold, Sentinel Diagnostics, Milan, Italy), only a small difference in diagnostic performance when the FIT was conducted either based on frozen stool samples (like in this study) or on fecal specimens directly collected with the recommended FIT tube from the whole bowel movement by the study participants [33]. Another limitation is that the analysis is based on a rather low number of CRC cases recruited in the screening setting (*n* = 16), which, on the other hand, is typical in a screening setting among average-risk participants of screening colonoscopy [15,22]. However, by including a separate group of CRC cases recruited in a clinical setting (*n* = 50), of whom 27 were also detected via a screening colonoscopy, more precise estimates of sensitivity were possible. We observed very similar sensitivity results for both groups of CRC, which, in addition, go in hand with the published results of other colonoscopy-controlled FIT-studies [15,22]. Furthermore, the estimates of specificity are based on a random selection of “only” 300 individuals out of over 1400 eligible participants without CRC and AA. Nevertheless, our specificity results go again in hand with results of other colonoscopy-controlled FIT-studies [15,22]. Therefore, we believe that this type of study design, with a targeted selection of all AN cases (*n* = 216), and a random selection of participants without AN (*n* = 300) seems to be a justified and efficient approach to save resources and capacities.

In conclusion, our study provides important, previously unavailable results regarding the combination of a variety of different FITs for CRC screening. We have shown that even FITs with different analytical characteristic correlate strongly across each other and detect in the majority of cases the same participants with AN (i.e., CRC or AA), and that combinations of different FITs do not lead to a relevant improvement in diagnostic performance. Therefore, FIT-based screening programs should consider selecting a FIT that fits best to their current CRC screening program based on factors like costs of tests, usability, laboratory requirements, simplicity of test analysis or ability of sample stabilization in the respective buffer-filled FIT tubes until test evaluation [31], and choose the positivity cutoff according to a defined target level of specificity (or overall positivity rate, which is highly correlated to specificity [15]) and available resources to follow up positive test results.

## 4. Materials and Methods

This article is following the STARD (Standards for Reporting of Diagnostic Accuracy) [34] and the FITTER (Fecal Immunochemical Tests for Hemoglobin Evaluation Reporting) [35] guidelines.

### 4.1. Study Design and Study Population

Details of the study design and study population have been provided in a previous report on the individual diagnostic performance of each of the nine different quantitative FITs [22]. Briefly, study participants from the BLITZ (Begleitende Evaluierung innovativer Testverfahren zur Darmkrebsfrüherkennung) study, an ongoing prospective study among participants of the German screening colonoscopy program were asked to collect a stool sample before starting bowel preparation for colonoscopy. Participants are informed and recruited at a preparatory visit in cooperating gastroenterology practices before screening colonoscopy.

To increase the number of CRC cases, which is typically low among average-risk individuals in true screening settings, an ancillary group of CRC cases from the DACHSplus study, which is an add-on study to the DACHS study (Darmkrebs: Chancen der Verhütung durch Screening), was included in a clinical setting. In the DACHSplus study, participants were asked to collect a stool sample before starting bowel preperation for surgery. Particpants were informed and recruited before initiating therapy.

Both studies have been approved by the Ethics committees of the University of Heidelberg and by the State Chambers of Physicians of Baden-Wuerttemberg, Rhineland-Palatinate and Hesse (BLITZ study (178/2005): 13 June 2005 and DACHS plus study (310/2001) 27 March 2006). Written informed consent was obtained from each study participant.

### 4.2. Selection of Study Participants

Figure 1 shows the flow diagram and exclusion criteria of the study participants. Study participants, who were recruited between 2005 and 2010 and provided stool samples, were considered for this project. From the BLITZ study, all eligible 216 individuals with CRC or AA (defined as adenoma with at least one of the following features: ≥1 cm in size, tubulovillous or villous components, or high-grade dysplasia) and 300 randomly selected participants without CRC and AA were included. From the DACHS plus study, a total of 50 CRC cases were included for ancillary analyses.

### 4.3. Sample and Data Collection

Study participants from both studies were asked to fill a 60 mL container with stool from a single bowel movement, without any dietary or medicinal restrictions, before starting bowel preparation for colonoscopy (BLITZ) or surgery (DACHS plus). Participants were asked to store the stool-filled container frozen or, if not possible, refrigerated. On the day of their colonoscopy appointment (BLITZ) or hospital admission (DACHS plus), the participants were asked to bring the frozen stool-filled container in a temperature-isolated manner to their gastroenterology practice or hospital. Upon receipt the containers were kept frozen at –20 °C and shipped on dry ice to the German Cancer Research Center (DKFZ) for final storage at –80 °C.

Screening colonoscopy was performed blinded to the test results. Colonoscopy and histology reports (BLITZ) as well as medical reports after surgery (DACHS plus) were collected from all study participants. Relevant information was extracted by two independent, trained research assistants who were blinded to the test results.

### 4.4. Fecal Immunochemical Test Analysis

Five laboratory-based tests (IDK Hb ELISA, Immundiagnostik, Bensheim, Germany; RIDASCREEN Hb, R-Biopharm, Darmstadt, Germany; CAREprime Hb, Alfresa Pharma, Osaka, Japan; OC Sensor, Eiken Chemical, Tokyo, Japan; and SENTiFIT-FOB Gold, Sentinel Diagnostics, Milan Italy) and four point-of-care tests (QuantOn Hem, Immundiagnostik, Bensheim, Germany; immoCARE-C, CARE diagnostica, Möllersdorf, Austria; Eurolyser FOB test, Eurolyser Diagnostica, Salzburg, Austria; and QuikRead go iFOBT, Orion Diagnostica, Espoo, Finland) were conducted in parallel. Characteristics of the nine different FITs are provided in Table 5.

Stool containers, from both studies, were blinded and put in random order before they were thawed for FIT analyses. Afterwards the stool samples were mixed to reduce potential heterogeneity of Hb concentration within the same bowel movement [32] as a potential source of variation of test measurements between the different FITs. A defined amount of stool was extracted using each manufacturer’s FIT-specific fecal sampling tube. Each FIT tube was a small vial, containing a notched stick for stool collection. After inserting the collection stick into three different areas of the stool sample, the notches of the stick were visually inspected for complete filling. Then the stick was pushed back into the vial, which is filled with a defined volume (range: 1.5–2.5 mL) of a FIT-specific preservation buffer to slow down any Hb degradation from sampling until test evaluation [31]. The vials had a tight entrance that wiped off excess stool from the stick, when the stick was pushed back into the vial, leaving only a defined mass of stool in the notches of the stick (range: 9.5–20 mg). The only exception was the immoCARE-C vial, where a supplied cardboard fork was used to wipe off excess stool from the stick before putting it back into the vial.

Afterwards the stool-filled FIT tubes were shaken on a vortexer so that the stool could move out of the notches of the stick into the preservation buffer to ensure optimal stabilization from sampling until test evaluation [31]. All tests were evaluated in parallel, automatically by the analytical instrument and in a one-time measurement by trained laboratory staff. Test calibrations and controls were performed on a regular basis according to the manufacturers´ instructions. Test results above the upper analytical limit were diluted and re-tested, if possible. Due to limited laboratory space and resources, five of the nine tests were evaluated externally, but under the same pre-analytical and analytical conditions: After vortexing, the FIT tubes were shipped to the following distributing and cooperating companies for blinded test evaluations on the next day:CARE diagnostica GmbH, Voerde, Germany (CAREprime Hb and immoCARE-C)Immundiagnostik AG, Bensheim, Germany (IDK Hb ELISA and QuantOn Hem)R‑Biopharm AG, Darmstadt, Germany (RIDASCREEN Hb).

The internal and external FIT tubes were stored at a median temperature of 21.5 °C (range: 20.0–24.0 °C) until blinded test analyses on the next day.

### 4.5. Statistical Analyses

Before starting the statistical analysis, all quantitative FIT measurements were converted to the same and comparable unit of µg Hb/g feces [36]. Sensitivities and specificities were calculated for single tests (*n* = 9) and for pairwise combinations of tests (*n* = 36) requiring either both tests (P++) or at least one test (P+) to be positive for defining a positive test result. In order to enhance comparability, these estimates were not only computed at original cutoffs (range: 2–17 µg Hb/g feces) and at a uniform cutoff (15 µg Hb/g feces), but also at cutoffs adjusted to yield the same specificity (96.7% and 93.0%, respectively). Sensitivities were calculated for detection of CRC (separetely for cases from the screening (*n* = 16) and clinical (*n* = 50) setting), AA (*n* = 200) and AN (*n* = 216, screening setting only), respectively, and specificities were computed for participants without AN. Screening colonoscopy results (gold-standard, BLITZ) and medical reports (DACHSplus) were used as reference standard. For one test (immoCARE-C) the analyses were based on one less AA case (*n* = 199), because of a missing FIT measurement. Due to the lower analytical limit of another test (QuikRead go iFOBT), the cutoff value could not be adjusted to yield a specificity below 96.7%, this test was excluded from the analysis at 93% specificity.

Median sensitivities and specificities with their range were computed for each single test and each of the various two-test combinations. Furthermore absolute differences in percent units between each test combination and the single test with the higher sensitivity and specificity, respectively, their medians and range were calculated (Table 1 and Table 2). In addition, specificities and sensitivities for AN (only particpants of screening colonoscopy, BLITZ) were graphically displayed at original cutoffs (range: 2–17 µg/g), at a uniform (15 µg/g) and at adjusted cutoffs, yielding equal specificities (Figure 2).

AUCs for detection of AN (only particpants of screening colonoscopy, BLITZ) were calculated for the nine single FITs and for their pairwise combinations, using logistic regression model. In addition, the AUC for AN was calculated by combining all nine FITs together.

Spearman rank correlation coefficients between the quantitative FIT measurements were computed for participants of screening colonscopy with CRC, participants with AA, participants without CRC and AA (No AN), and for all included study participants of screening colonscopy (Total), respectively across all 36 two-test comparisons (Table 3). Similarly, Cohen’s kappa for agreement of test classifications as positive or negative according to original cutoffs and to adjusted cutoffs, yielding the same specificity of 96.7% were calculated (Table 4).

All statistical analyses were conducted using SAS Enterprise Guide, version 6.1 (SAS Institute Inc., Cary, CA, USA).

## 5. Conclusions

Different quantitative FITs from diverse manufacturers yield strongly correlated results and are able to detect in the majority of cases the same participants with AN. As a result, no relevant gain in diagnostic performance was found by the combination of different FITs.

## Figures and Tables

**Figure 1 cancers-11-00120-f001:**
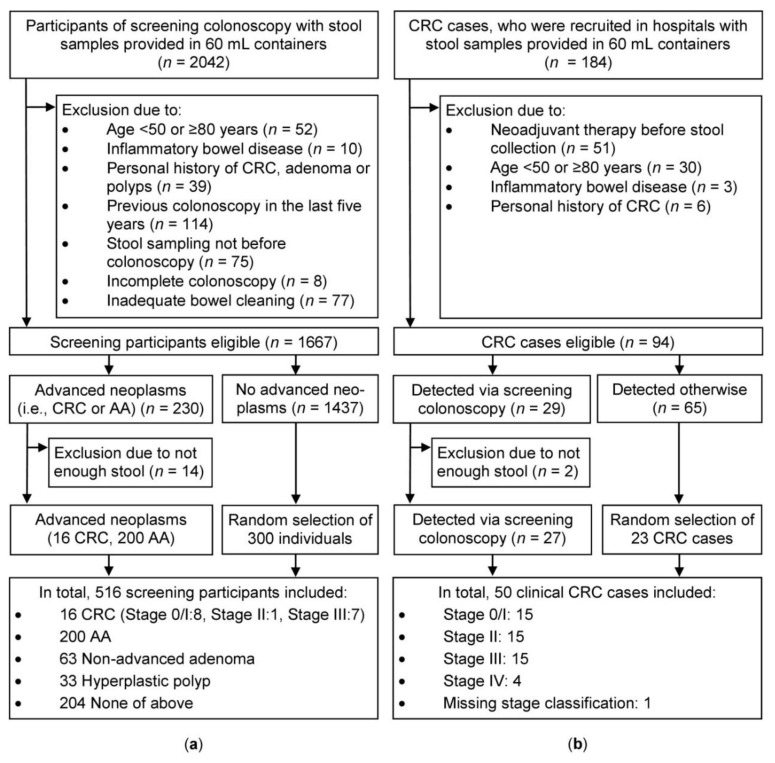
Flow diagram for selection of study population (**a**) from the screening setting (Begleitende Evaluierung innovativer Testverfahren zur Darmkrebsfrüherkennung (BLITZ) study, left side) and (**b**) from the clinical setting (Darmkrebs: Chancen der Verhütung durch Screening (DACHS) plus study, right side). AA: Advanced adenoma; CRC: Colorectal cancer; CRC stage classification according to the Union for International Cancer Control (UICC) (7th edition).

**Figure 2 cancers-11-00120-f002:**
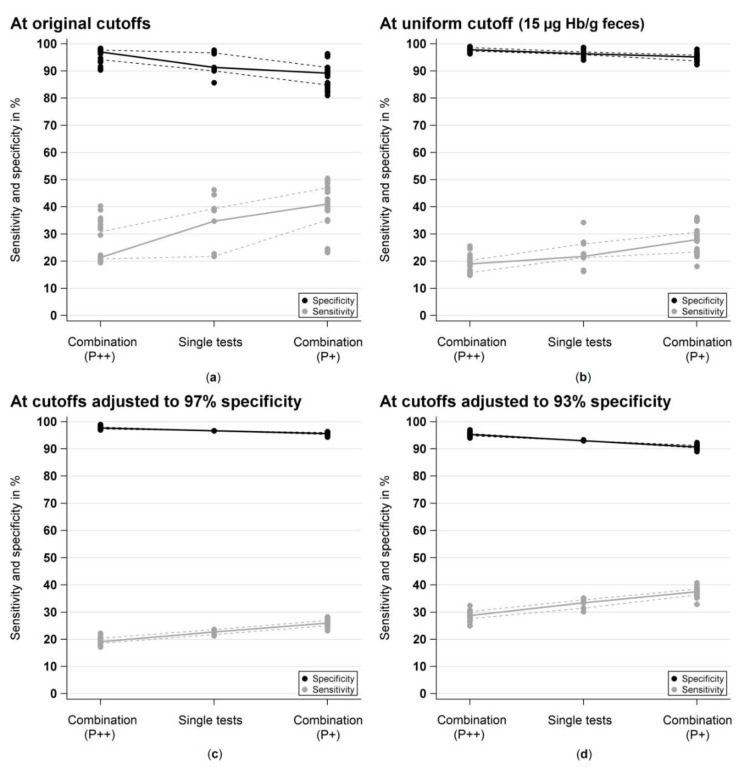
Sensitivities and specificities for detection of advanced neoplasm (AN; i.e., colorectal cancer and advanced adenoma) at (**a**) original, (**b**) uniform and (**c** and **d**) adjusted cutoff values for single tests and for their pairwise combinations, respectively. Solid line: Connected medians; Dashed line: Connected interquartile ranges.

**Table 1 cancers-11-00120-t001:** Sensitivity and specificity of single tests and of two-test combinations assuming positivity if both tests are positive (P++).

Cutoff	Test(s)	Metric	Participants of Screening Colonoscopy (*n* = 516)				Clinical CRC Cases
			Sensitivity (%)			Specificity (%)	Sensitivity (%)
			CRC (*n* = 16)	AA (*n* = 200)	AN (*n* = 216)	No AN (*n* = 300)	CRC (*n* = 50)
Original cutoff (range: 2–17 µg/g)	Single tests (*n* = 9)	Median	81.3	31.0	34.7	91.3	74.0
		Range	62.5 to 81.3	18.0 to 43.5	21.8 to 46.3	85.7 to 97.7	64.0 to 84.0
	Combinations (*n* = 36)	Median	68.8	18.0	21.4	97.0	68.0
		Range	62.5 to 81.3	16.0 to 37.5	19.4 to 40.3	90.3 to 98.3	62.0 to 80.0
	Difference (*n* = 36) ^†^	Median	−12.5	−14.0	−13.7	0.7	−8.0
		Range	−18.8 to 0.0	−26.5 to −1.5	−25.5 to −1.4	0.0 to 4.7	−22.0 to −2.0
Uniform Cutoff ( = 15 µg/g)	Single tests (*n* = 9)	Median	68.8	18.5	21.8	96.3	70.0
		Range	56.3 to 81.3	13.0 to 30.5	16.2 to 34.3	94.0 to 98.7	56.0 to 76.0
	Combinations (*n* = 36)	Median	62.5	15.0	19.0	97.8	68.0
		Range	56.3 to 75.0	11.5 to 21.6	14.8 to 25.6	96.3 to 99.0	52.0 to 72.0
	Difference (*n* = 36) ^†^	Median	−9.4	−7.0	−7.0	0.7	−6.0
		Range	−25.0 to 0.0	−18.0 to −1.5	−18.5 to −1.9	0.0 to 1.7	−20.0 to −0.0
Cutoffs adjusted to 96.7% specificity (range: 6–30 µg/g)	Single tests (*n* = 9)	Median	68.8	19.0	22.7	96.7	68.0
		Range	62.5 to 75.0	17.5 to 20.1	21.3 to 23.6	96.7 to 96.7	64.0 to 74.0
	Combinations (*n* = 36)	Median	62.5	15.5	19.1	97.7	66.0
		Range	62.5 to 68.8	13.5 to 18.5	17.1 to 22.2	97.0 to 99.0	62.0 to 70.0
	Difference (*n* = 36) ^†^	Median	−6.3	-4.0	−4.2	1.0	−4.0
		Range	−12.5 to 0.0	−6.0 to −1.5	−5.6 to −1.4	0.3 to 2.3	−12.0 to 0.0
Cutoffs adjusted to 93.0% specificity (range: 2–13 µg/g) ^‡^	Single tests (*n* = 8)	Median	78.1	29.6	33.4	93.0	74.0
		Range	68.8 to 81.3	26.5 to 31.5	30.1 to 35.2	93.0 to 93.3 ^§^	72.0 to 80.0
	Combinations (*n* = 28)	Median	75.0	25.3	28.8	95.3	71.0
		Range	68.8 to 81.3	21.0 to 28.5	25.0 to 32.4	94.0 to 97.0	70.0 to 74.0
	Difference (*n* = 28) ^†^	Median	−6.3	−5.0	−5.3	2.3	−4.0
		Range	−12.5 to 0.0	−9.5 to −3.0	−9.3 to −2.8	0.7 to 4.0	−10.0 to 0.0

AA: Advanced adenoma; AN: Advanced neoplasm; CRC: Colorectal cancer; No AN: Participants without AN; **^†^**: Difference compared to the respective single test with the higher sensitivity and specificity, respectively; **^‡^**: One test (QuikRead go iFOBT, Orion Diagnostica, Espoo, Finland) was excluded, because the cutoff value could not be adjusted to yield a specificity lower than 96.7%; **^§^**: For one of the remaining eight tests (SENTiFIT-FOB Gold, Sentinel Diagnostics, Milan, Italy) the cutoff value could not be decreased further to yield a specificity lower than 93.3%.

**Table 2 cancers-11-00120-t002:** Sensitivity and specificity of single tests and of two-test combinations assuming positivity if at least one test is positive (P+).

Cutoff	Test(s)	Metric	Participants of Screening Colonscopy (*n* = 516)				Clinical CRC Cases
			Sensitivity (%)			Specificity (%)	Sensitivity (%)
			CRC (*n* = 16)	AA (*n* = 200)	AN (*n* = 216)	No AN (*n* = 300)	CRC (*n* = 50)
Original cutoff (range: 2–17 µg/g)	Single tests (*n* = 9)	Median	81.3	31.0	34.7	91.3	74.0
		Range	62.5 to 81.3	18.0 to 43.5	21.8 to 46.3	85.7 to 97.7	64.0 to 84.0
	Combinations (*n* = 36)	Median	81.3	37.8	41.1	89.2	78.0
		Range	62.5 to 87.5	19.5 to 47.5	23.2 to 50.5	81.0 to 96.3	68.0 to 88.0
	Difference (*n* = 36) ^†^	Median	0.0	1.0	1.2	−6.7	0.0
		Range	0.0 to 6.3	0.0 to 5.0	0.0 to 5.1	−12.7 to −1.3	0.0 to 4.0
Uniform Cutoff (= 15 µg/g)	Single tests (*n* = 9)	Median	68.8	18.5	21.8	96.3	70.0
		Range	56.3 to 81.3	13.0 to 30.5	16.2 to 34.3	94.0 to 98.7	56.0 to 76.0
	Combinations (*n* = 36)	Median	75.0	24.1	27.9	95.2	72.0
		Range	56.3 to 81.3	15.0 to 32.5	18.1 to 36.1	92.3 to 98.0	68.0 to 78.0
	Difference (*n* = 36) ^†^	Median	0.0	1.5	1.4	−2.3	0.0
		Range	0.0 to 6.3	0.0 to 4.0	0.0 to 4.2	−4.7 to −0.7	0.0 to 4.0
Cutoffs adjusted to 96.7% specificity (range: 6–30 µg/g)	Single tests (*n* = 9)	Median	68.8	19.0	22.7	96.7	68.0
		Range	62.5 to 75.0	17.5 to 20.1	21.3 to 23.6	96.7 to 96.7	64.0 to 74.0
	Combinations (n = 36)	Median	68.8	22.5	25.9	95.7	70.0
		Range	62.5 to 75.0	19.5 to 24.6	23.2 to 28.2	94.3 to 96.3	68.0 to 76.0
	Difference (*n* = 36) ^†^	Median	0.0	3.0	2.8	−1.0	0.0
		Range	0.0 to 0.0	1.0 to 5.0	0.9 to 4.7	−2.3 to −0.3	0.0 to 2.0
Cutoffs adjusted to 93.0% specificity (range: 2–13 µg/g) ^‡^	Single tests (*n* = 8)	Median	78.1	29.6	33.4	93.0	74.0
		Range	68.8 to 81.3	26.5 to 31.5	30.1 to 35.2	93.0 to 93.3 ^§^	72.0 to 80.0
	Combinations (*n* = 28)	Median	81.3	34.0	37.5	90.7	76.0
		Range	75.0 to 81.3	29.5 to 37.5	32.9 to 40.7	89.0 to 92.3	74.0 to 84.0
	Difference (*n* = 28) ^†^	Median	0.0	3.2	3.0	−2.3	2.0
		Range	0.0 to 6.3	1.0 to 7.0	1.4 to 6.5	−4.0 to −1.0	0.0 to 4.0

AA: Advanced adenoma; AN: Advanced neoplasm; CRC: Colorectal cancer; No AN: Participants without AN; **^†^**: Difference compared to the respective single test with the higher sensitivity and specificity, respectively; **^‡^**: One test (QuikRead go iFOBT, Orion Diagnostica, Espoo, Finland) was excluded, because the cutoff value could not be adjusted to yield a specificity lower than 96.7%; **^§^**: For one of the remaining eight tests (SENTiFIT-FOB Gold, Sentinel Diagnostics, Milan, Italy) the cutoff value could not be decreased further to yield a specificity lower than 93.3%.

**Table 3 cancers-11-00120-t003:** Spearman rank correlation coefficients between quantitative test measurements (µg Hb/g feces).

FIT Brand	Ridascreen Hb	QuantOn Hem	immoCARE‑C ^†^	CAREprime Hb	Eurolyser FOB test	OC Sensor	QuikRead go iFOBT	SENTiFIT-FOB Gold
IDK Hb ELISA	**CRC 0.90**	**CRC 0.89**	CRC 0.84	CRC 0.83	CRC 0.91	CRC 0.85	CRC 0.92	CRC 0.83
	**AA 0.95**	**AA 0.89**	AA 0.87	AA 0.80	AA 0.74	AA 0.81	AA 0.62	AA 0.75
	**No AN 0.81**	**No AN 0.62**	No AN 0.50	No AN 0.48	No AN 0.26	No AN 0.44	No AN 0.28	No AN 0.31
	**Total 0.90**	**Total 0.78**	Total 0.74	Total 0.68	Total 0.58	Total 0.66	Total 0.51	Total 0.59
Ridascreen Hb		**CRC 0.89**	CRC 0.90	CRC 0.90	CRC 0.90	CRC 0.90	CRC 0.88	CRC 0.89
		**AA 0.85**	AA 0.87	AA 0.83	AA 0.77	AA 0.81	AA 0.61	AA 0.76
		**No AN 0.62**	No AN 0.57	No AN 0.52	No AN 0.32	No AN 0.50	No AN 0.34	No AN 0.40
		**Total 0.78**	Total 0.77	Total 0.72	Total 0.64	Total 0.70	Total 0.55	Total 0.65
QuantOn Hem			CRC 0.91	CRC 0.91	CRC 0.89	CRC 0.89	CRC 0.93	CRC 0.92
			AA 0.77	AA 0.72	AA 0.68	AA 0.71	AA 0.60	AA 0.69
			No AN 0.38	No AN 0.34	No AN 0.24	No AN 0.22	No AN 0.27	No AN 0.28
			Total 0.63	Total 0.58	Total 0.56	Total 0.53	Total 0.52	Total 0.57
immoCARE‑C ^†^				CRC 0.92	CRC 0.89	CRC 0.83	CRC 0.88	CRC 0.89
				AA 0.80	AA 0.79	AA 0.77	AA 0.63	AA 0.76
				No AN 0.47	No AN 0.41	No AN 0.51	No AN 0.32	No AN 0.45
				Total 0.68	Total 0.67	Total 0.68	Total 0.54	Total 0.65
CAREprime Hb					CRC 0.81	CRC 0.96	CRC 0.89	CRC 0.98
					AA 0.79	AA 0.82	AA 0.66	AA 0.77
					No AN 0.40	No AN 0.54	No AN 0.31	No AN 0.43
					Total 0.65	Total 0.71	Total 0.53	Total 0.65
Eurolyser FOB test						CRC 0.83	CRC 0.85	**CRC 0.82**
						AA 0.81	AA 0.80	**AA 0.92**
						No AN 0.45	No AN 0.54	**No AN 0.79**
						Total 0.69	Total 0.75	**Total 0.89**
OC Sensor							CRC 0.87	CRC 0.96
							AA 0.67	AA 0.80
							No AN 0.35	No AN 0.48
							Total 0.57	Total 0.69
QuikRead go iFOBT								CRC 0.92
								AA 0.80
								No AN 0.57
								Total 0.77

AA: Advanced adenoma (*n* = 200); CRC: Colorectal cancer (*n* = 16); Hb: Hemoglobin; No AN: Participants without advanced neoplasms (AN, i.e., CRC or AA) (*n* = 300); Total: All study participants of screening colonoscopy (*n* = 516); **^†^**: Analyses based on one less AA case (*n* = 199); bold type: Strong correlation across all study groups (r_s_ > 0.60).

**Table 4 cancers-11-00120-t004:** Cohen’s kappa for agreement of test classification as positive or negative at original cutoffs (values above diagonal) and at adjusted cutoffs, yielding 96.7% specificity (values below diagonal), respectively.

FIT Brand	IDK Hb ELISA	Ridascreen Hb	QuantOn Hem	immoCARE‑C ^†^	CAREprime Hb	Eurolyser FOB Test	OC Sensor	QuikRead go iFOBT	SENTiFIT-FOB Gold
**IDK Hb ELISA**		**CRC 1.00**	CRC 0.59	CRC 1.00	CRC 1.00	CRC 0.56	CRC 0.67	CRC 0.56	CRC 0.67
		**AA 0.82**	AA 0.80	AA 0.71	AA 0.63	AA 0.44	AA 0.42	AA 0.41	AA 0.40
		**No AN 0.76**	No AN 0.62	No AN 0.71	No AN 0.53	No AN 0.19	No AN 0.25	No AN 0.30	No AN 0.29
		**Total 0.83**	Total 0.76	Total 0.76	Total 0.66	Total 0.44	Total 0.46	Total 0.45	Total 0.45
**Ridascreen Hb**	**CRC 0.86**		CRC 0.59	**CRC 1.00**	**CRC 1.00**	CRC 0.56	CRC 0.67	CRC 0.56	CRC 0.67
	**AA 0.82**		AA 0.70	**AA 0.81**	**AA 0.75**	AA 0.53	AA 0.54	AA 0.48	AA 0.51
	**No AN 0.90**		No AN 0.51	**No AN 0.85**	**No AN 0.72**	No AN 0.29	No AN 0.38	No AN 0.45	No AN 0.43
	**Total 0.86**		Total 0.68	**Total 0.86**	**Total 0.79**	Total 0.54	Total 0.57	Total 0.54	Total 0.57
**QuantOn Hem**	**CRC 0.85**	**CRC 0.71**		CRC 0.59	CRC 0.59	CRC 0.56	CRC 0.67	CRC 0.56	CRC 0.67
	**AA 0.76**	**AA 0.79**		AA 0.66	AA 0.56	AA 0.42	AA 0.43	AA 0.44	AA 0.43
	**No AN 0.69**	**No AN 0.59**		No AN 0.46	No AN 0.40	No AN 0.15	No AN 0.25	No AN 0.26	No AN 0.25
	**Total 0.79**	**Total 0.78**		Total 0.64	Total 0.57	Total 0.41	Total 0.46	Total 0.46	Total 0.45
**immoCARE‑C †**	**CRC 0.86**	**CRC 1.00**	CRC 0.71		**CRC 1.00**	CRC 0.56	CRC 0.67	CRC 0.56	CRC 0.67
	**AA 0.80**	**AA 0.71**	AA 0.69		**AA 0.80**	AA 0.57	AA 0.58	AA 0.54	AA 0.55
	**No AN 0.79**	**No AN 0.69**	No AN 0.48		**No AN 0.69**	No AN 0.33	No AN 0.35	No AN 0.42	No AN 0.41
	**Total 0.83**	**Total 0.77**	Total 0.71		**Total 0.80**	Total 0.57	Total 0.59	Total 0.57	Total 0.58
**CAREprime Hb**	**CRC 1.00**	**CRC 0.86**	CRC 0.85	**CRC 0.86**		CRC 0.56	CRC 0.67	CRC 0.56	CRC 0.67
		**AA 0.85**	**AA 0.78**	AA 0.73		**AA 0.78**	AA 0.67	AA 0.66	AA 0.65	AA 0.66
		**No AN 0.79**	**No AN 0.79**	No AN 0.48		**No AN 0.69**	No AN 0.49	No AN 0.40	No AN 0.53	No AN 0.57
		**Total 0.87**	**Total 0.82**	Total 0.74		**Total 0.80**	Total 0.67	Total 0.65	Total 0.66	Total 0.68
**Eurolyser FOB test**	CRC 1.00	CRC 0.86	CRC 0.85	**CRC 0.86**	**CRC 1.00**		**CRC 0.86**	CRC 1.00	**CRC 0.86**
		AA 0.79	AA 0.75	AA 0.66	**AA 0.72**		**AA 0.91**	**AA 0.89**	AA 0.84	**AA 0.85**
		No AN 0.59	No AN 0.59	No AN 0.28	**No AN 0.69**		**No AN 0.69**	**No AN 0.61**	No AN 0.51	**No AN 0.69**
		Total 0.80	Total 0.76	Total 0.66	**Total 0.77**		**Total 0.89**	**Total 0.86**	Total 0.81	**Total 0.84**
**OC Sensor**	**CRC 1.00**	**CRC 0.86**	CRC 0.85	CRC 0.86	**CRC 1.00**	CRC 1.00		CRC 0.86	**CRC 1.00**
		**AA 0.82**	**AA 0.78**	AA 0.69	AA 0.72	**AA 0.88**		AA 0.81	AA 0.85	**AA 0.86**
		**No AN 0.79**	**No AN 0.90**	No AN 0.48	No AN 0.59	**No AN 0.90**		No AN 0.59	No AN 0.58	**No AN 0.66**
		**Total 0.86**	**Total 0.84**	Total 0.72	Total 0.75	**Total 0.91**		Total 0.81	Total 0.83	**Total 0.86**
**QuikRead go iFOBT**	**CRC 0.86**	CRC 1.00	CRC 0.71	CRC 1.00	**CRC 0.86**	CRC 0.86	**CRC 0.86**		**CRC 0.86**
		**AA 0.80**	AA 0.72	AA 0.67	AA 0.69	**AA 0.86**	AA 0.86		**AA 0.79**	**AA 0.92**
		**No AN 0.69**	No AN 0.59	No AN 0.59	No AN 0.59	**No AN 0.79**	No AN 0.48		**No AN 0.69**	**No AN 0.65**
		**Total 0.81**	Total 0.76	Total 0.71	Total 0.74	**Total 0.87**	Total 0.81		**Total 0.81**	**Total 0.87**
**SENTiFIT- FOB Gold**	**CRC 1.00**	**CRC 0.86**	CRC 0.85	**CRC 0.86**	**CRC 1.00**	**CRC 1.00**	**CRC 1.00**	**CRC 0.86**	
	**AA 0.81**	**AA 0.73**	AA 0.68	**AA 0.67**	**AA 0.87**	**AA 0.87**	**AA 0.84**	**AA 0.92**	
	**No AN 0.69**	**No AN 0.69**	No AN 0.38	**No AN 0.69**	**No AN 0.90**	**No AN 0.79**	**No AN 0.79**	**No AN 0.69**	
	**Total 0.83**	**Total 0.77**	Total 0.69	**Total 0.74**	**Total 0.90**	**Total 0.89**	**Total 0.87**	**Total 0.88**	

AA: Advanced adenoma (*n* = 200); CRC: Colorectal cancer (*n* = 16); Hb: Hemoglobin; No AN: Participants without advanced neoplasms (AN, i.e., CRC or AA) (*n* = 300); Total: All study participants of screening colonoscopy (*n* = 516); **^†^**: Analyses based on one less AA case; bold type: Strong correlation across all study groups (kappa > 0.60).

**Table 5 cancers-11-00120-t005:** Characteristics of the included quantitative FITs.

FIT Brand	Manufacturer, City, Country	Fecal Sampling Tube (Fecal Mass/Buffer Volume)	Analytical Instrument	Analytical Reading Technique	Analytical Working Range (µg Hb/g feces)	Original Cutoff (µg Hb/g feces)
IDK Hb ELISA	Immundiagnostik, Bensheim, Germany	IDK Extract (15 mg/1.5 mL)	DSX by Dynex Technologies	Enzyme-linked immunosorbent assay (ELISA)	0.086 to 50	2.00
RIDASCREEN Hb	R-Biopharm, Darmstadt, Germany	RIDA TUBE Hb (10 mg/2.5 mL)	DSX by Dynex Technologies	Enzyme-linked immunosorbent assay (ELISA)	0.65 to 50	8.00
QuantOn Hem	Immundiagnostik, Bensheim, Germany	QuantOn Hem TUBE (15 mg/1.5 mL)	QuantOn Hem test cassette & Smartphone ^†^	Immunoaffinity chromatography & Photometry	0.3 to 100	3.70
immoCARE-C	CARE diagnostica, Möllersdorf, Austria	Sample Collection Tube (20 mg/2.5 mL)	immoCARE‑C test cassette & CAREcube	Immunoaffinity chromatography & Photometry	3.75 to 250	6.25
CAREprime Hb	Alfresa Pharma, Osaka, Japan	Specimen Collection Container A (9.5 mg/1.9 mL)	CAREprime	Immunoturbidimetry	1.6 to 240	6.30
Eurolyser FOB test	Eurolyser Diagnostica, Salzburg, Austria	Eurolyser FOB Sample Collector (19.9 mg/1.6 mL)	Eurolyser CUBE	Immunoturbidimetry	2.01 to 80.4	8.04
OC Sensor	Eiken Chemical, Tokyo, Japan	OC Auto-Sampling Bottle 3 (10 mg/2.0 mL)	OC Sensor io	Immunoturbidimetry	10 to 200	10.0
QuikRead go iFOBT	Orion Diagnostica, Espoo, Finland	QuikRead go iFOBT Sampling Set (10 mg/2.0 mL)	QuikRead go	Immunoturbidimetry	15 to 200	15.0
SENTiFIT‑FOB Gold	Sentinel Diagnostics, Milan, Italy	SENTiFIT pierceTube (10 mg/1.7 mL)	SENTiFIT 270 analyzer	Immunoturbidimetry	1.7 to 129.88	17.0

FIT: Fecal immunochemical test; Hb: Hemoglobin; **^†^**: iPhone 6s with special test evaluation software (designed by the manufacturer) was used for this study.

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
