# Peer review of "Combination of Different Fecal Immunochemical Tests in Colorectal Cancer Screening: Any Gain in Diagnostic Performance?"

_cancers, 2019, doi:10.3390/cancers11010120_

Round 1
Reviewer 1 Report
The authors are to be congratulated on accomplishing a valuable study.
I believe that my comment would further improve the quality of your study.
As you show in Table 4, the sensitivity of FITs for advanced adenoma is lower than that for CRC. Your cohort included only 16 colorectal cancers (CRCs) among 216 advanced neoplasms, which may led to a low sensitivity across the FITs. A higher sensitivity is required for CRC screening at the cost of lowering the specificity in the United States (Lin JS, Piper MA, Perdue LA, Rutter C, Webber EM, O’Connor E, Smith N, Whitlock EP. Screening for Colorectal Cancer: A Systematic Review for the U.S. Preventive Services Task Force. Evidence Synthesis No. 135. AHRQ Publication No. 14-05203-EF-1. Rockville, MD: Agency for Healthcare Research and Quality; 2016). Please discuss about the reproducibility of your findings in CRC screening in the real world.
Author Response
Dear Reviewer,
thank you very much for your constructive review.
Please find attached our Point-by-Point Response.
Best wishes,
Anton Gies

Reviewer 2 Report
Overview:
In the manuscript “Combination of different fecal immunochemical tests in colorectal cancer screening: Any gain in diagnostic performance?” by Gies et al., the authors aim to explore the potential for enhancing diagnostic performance by pairwise combinations of nine different quantitative fecal immunochemical tests based on stool samples. The manuscript although straightforward, well written and the analyses are generally properly executed, has some flaws that concern this reviewer.
Concerns:
1. One concern with the experimental analyses is that the authors have concluded based on the data collected from a relatively small number of patients. According to this reviewer, to make the analysis more relevant, the authors should try to include more patients. This will help in addressing the reproducibility issue.
2. The overall format of the manuscript, including the figures, should be corrected. Additionally, there are a few grammatical and spelling errors that need to be corrected.
3. The reference list needs formatting.
Author Response
Dear Reviwer,
thank your very much for your constructive Review.
Please find attached our Point-by-Point Response.
Best wishes,
Anton Gies

Round 2
Reviewer 2 Report
The authors have addressed all the concerns raised by this reviewer. The revised manuscript seems to be suitable for publication.